Constructing a comprehensive gene co-expression based interactome in Bos taurus

Chen Yan 1
Liu Yining 2
Du Min 3
Zhang Wengang 1
Xu Ling 1
Gao Xue 1
Zhang Lupei 1
Gao Huijiang 1
Xu Lingyang 1
Li Junya jl1@iascaas.net.cn 1
Zhao Min mzhao@usc.edu.au 4
1 Innovation Team of Cattle Genetics and Breeding, Institute of Animal Science, Chinese Academy of Agricultural Sciences , Beijing , China
2 The School of Public Health, Institute for Chemical Carcinogenesis, Guangzhou Medical University , Guangzhou , China
3 Department of Animal Science, Washington State University , Pullman, WA , United States of America
4 School of Engineering, Faculty of Science, Health, Education and Engineering, University of the Sunshine Coast , Queensland , Australia
Loor Juan
Electronic publication date: 2017 Dec 4
Publication date: 2017
Volume: 5
Electronic Location ID: e4107
Received 2017 Aug 18; Accepted 2017 Nov 8
Copyright: ©2017 Chen et al.
Copyright year: 2017
Copyright holder: Chen et al.
License: This is an open access article distributed under the terms of the Creative Commons Attribution License, which permits unrestricted use, distribution, reproduction and adaptation in any medium and for any purpose provided that it is properly attributed. For attribution, the original author(s), title, publication source (PeerJ) and either DOI or URL of the article must be cited.
License URL: https://creativecommons.org/licenses/by/4.0/

Keywords: Co-expression, WGCNA, Network, Systems biology, Functional enrichment, Bos taurus

Funding: National Natural Science Foundation of China 31402039 Chinese Academy of Agricultural Sciences cxgc-ias-03 China’s Agriculture and Finance Ministries CARS-37 This work was supported by the National Natural Science Foundation of China (31402039), the start-up grant to Dr. Min Zhao, China Scholarship Council (CSC), Cattle Breeding Innovative Research Team of Chinese Academy of Agricultural Sciences (cxgc-ias-03), and National Beef Cattle Industrial Technology System of China’s Agriculture and Finance Ministries (CARS-37). The funders had no role in study design, data collection and analysis, decision to publish, or preparation of the manuscript.

==============================
Integrating genomic information into cattle breeding is an important approach to exploring genotype-phenotype relationships for complex traits related to diary and meat production. To assist with genomic-based selection, a reference map of interactome is needed to fully understand and identify the functional relevant genes. To this end, we constructed a co-expression analysis of 92 tissues and this represents the systematic exploration of gene-gene relationship in Bos taurus. By using robust WGCNA (Weighted Gene Correlation Network Analysis), we described the gene co-expression network of 5,000 protein-coding genes with majority variations in expression across 92 tissues. Further module identifications found 55 highly organized functional clusters representing diverse cellular activities. To demonstrate the re-use of our interaction for functional genomics analysis, we extracted a sub-network associated with DNA binding genes in Bos taurus. The subnetwork was enriched within regulation of transcription from RNA polymerase II promoter representing central cellular functions. In addition, we identified 28 novel linker genes associated with more than 100 DNA binding genes. Our WGCNA-based co-expression network reconstruction will be a valuable resource for exploring the molecular mechanisms of incompletely characterized proteins and for elucidating larger-scale patterns of functional modulization in the Bos taurus genome.

Introduction

As the importance in dairy and beef production, the genome of the domestic cattle, Bos taurus, was sequenced in 2009 using hierarchical and whole-genome shotgun sequencing strategy (Zimin et al., 2009). To associate the genetic variation with phenotypes, the first phase of the 1,000 bull genomes project was started to sequence 234 ancestor bulls (Daetwyler et al., 2014). Although more and more efforts for genetic improvement of production efficiency and quality in cattle, the majority of previous studies focused on single gene-based genetic breeding (Barabasi & Oltvai, 2004). However, most of production traits are complex traits involving multiple genes. The recent development of systems biology-based approach was promising to explore the genome and gene-gene interactions in a global view to understand molecular mechanisms underlying complex traits (Zhao, Kong & Qu, 2014).

A gene-based interactome is the complete set of gene-gene interactions in a particular cell (Barabasi & Oltvai, 2004) and these could be direct physical interactions among molecules as well as indirect interactions among genes (such as gene co-expression). The understanding of interactomes are important in systems biology-based studies as they provide a global view of all the possible molecular interactions that a protein can influence (Barabasi & Oltvai, 2004). Because of the lack of interactome in Bos taurus, the network-based data mining approach are not able to apply to functional discovery for any interesting genes associated with complex traits (Elsik et al., 2016). Recently, a large-scale gene co-expression for multiple cattle species and conditions were conducted (Beiki et al., 2016). However, this analysis is based on microarray data. With the high-throughput deep sequencing platform, cumulative expression data across multiple tissues in cattle are now publicly available and may promote a much better understanding of gene-gene interaction with more accurate data (Elsik et al., 2016).

In this study, we aim to provide a comprehensive gene expression-based interactome to explore the functional relevant genes for the most important cattle, Bos taurus. To this aim, we built a general framework of co-expression based interactome for Bos taurus through integrating expression profiles from 92 tissues from bovine genome database (BGD) (Elsik et al., 2016). In this study, we utilized an established network-based approach, Weighted Gene Co-Expression Network Analysis (WGCNA) (Langfelder & Horvath, 2008), to further identify and characterize a number of functional modules. To demonstrate our reconstructed interactome could provide a new approach for network-based data mining of cattle genetics data, we focused on the DNA-binding genes in Bos taurus and extracted a DNA-binding regulatory network.

Materials & Methods

The gene expression data in 92 tissues from bovine genome database

To characterized the gene expression in multiple tissues, we chose the gene expression dataset based on the criteria: (i) it covered the most comprehensive tissue types for Bos taurus; (ii) it was based on next-generation sequencing, which is more accurate than array-based quantification; (iii) it was generated in consistent and high-quality way. In the current GEO or ArrayExpress database, some of other datasets (like E-MTAB-2596) only have limited tissues. Therefore, we adopted gene expression data from 92 different tissues from the individual of the reference genome in the bovine genome database (BGD) (Elsik et al., 2016). By using RNAseq sequencing and mapping to the reference genome (University of Maryland version 3.1), all the genes in Bos taurus genome was quantified using the FPKM (Fragments Per Kilobase of transcript per Million mapped reads). All the FPKM were further normalized for each expression dataset by using cuffquant and cuffnorm to avoid any batch effects across samples. By using Intermine Web Services API of BovineMine (part of BGD), we downloaded all the normalized FPKM values of the 92 tissues. To further build the co-expression network based on high-quality data, we first removed those non-informative genes with FPKMs in 46 or less tissue samples. After the initial filtering, a list of 13,405 genes with FPKMs were subject to WGCNA analysis.

Reconstruction of a scale-free co-expression network from 92 cattle tissues using WGCNA

Network-based data mining is used to explore the behavior of all the gene-gene interactions. However, there is limited information about cattle in the gene-gene interaction database and, for instance, BioGrid (Chatr-Aryamontri et al., 2017) contains only 102 interaction pairs for cattle. To overcome this shortcoming, we used the co-expression network approach to reconstruct the functional interactome for cattle. Based on comprehensive transcriptomes with 92 tissue samples covering the majority tissue types in Bos taurus, we built and mined the gene co-expression network using the WGCNA analysis. WGCNA is a R package to construct gene co-expression networks. By using the package, we first built similarity matrix between all the gene pairs using bi-weight mid-correlation based on normalized FPKMs (Zheng et al., 2014).

Using 19,064 genes with expression values, we ran a quality control step and removed those genes without expression values in more than half of 92 tissues. This provided a list of 13,405 genes with expression across 92 tissues. However, a large number of these genes did not have expression variations between samples. Therefore, the data set with 13,405 gene expressions was processed further by focusing on the 5,000 most variant genes (Table S1). The remaining 8,405 genes, which showed no or very low changes in expression between samples, were not used for WGCNA analysis. The variability of gene expression data across the 92 samples was measured using a robust method called median absolute deviation (MAD). The 5,000 most variant genes were used for analysis in other WGCNA studies (De Jong et al., 2012).

To build a scale-free network, we run a parameter analysis (Fig. 1). Briefly, an adjacency function in WGCNA was used to weight between different genes in the hypothesis of following a power law. In detail, the correlation data were transformed to adjacency matrix using the formula: aij = (Sij, β) = |Sij|β. In the formula, the β represent the exponential parameter for power law distribution. Normally, the β was used to characterize the likeness to a scale-free network. In our data, the co-expression for a pair of gene represent a connection between two genes. In general, the number of connection of all the genes in a scale-free network follow a power law distribution P(k) ∼ k−β. The P(k) in our co-expressing network indicates the probability that a gene is co-expressed with k other genes. By setting the criterion that the co-efficiency of log(k) and log(p(k)) is greater than 0.8, we checked all the possible β values. As shown in Fig. 1A, we changed the β value step by step to identify the optimal value that the average connectivity of the network is smooth. The β = 4 was finally determined based on the diagnosis chart and the average number of co-expressed genes in the final network was 80 (Fig. 1B). Using this information, we reconstructed the first and most co-expression network in Bos taurus genome across 92 tissue samples representing the majority of tissue types; this will provide a basis for network-based data mining in Bos taurus genetics and genomics studies.

Figure 1 Determination of power Beta value based on the adjacency matrix using the weighted gene correlation network analysis (WGCNA).

The adjacency matrix from co-expression data was weighted by the power of the correlation data between different genes; i.e., aij = |Sij|β. The weighted parameter power Beta value was determined by the scale-free topology criterion. To ensure that the average connectivity of the network is smooth, we chose β = 4 based on both chart: (A) for topology fitting results and (B) for mean connectivity.

The identification of functional modules

To further identify functional modules in our reconstructed co-expression network with 5,000 genes, the adjacency matrix was further transformed to topological overlap matrix (TOM) using WGCNA package. The hierarchical clustering on all the genes were performed to generate a dendrogram. By using dynamic tree cutting, the functional clusters (modules) were obtained from the constructed gene dendrogram. In detail, the cutreeDynamic function in WGCNA package was used to identify the larger module with minimum size of 30 genes as possible. By setting parameter deepSplit from 0 to 4 for the tree cutting, we found the optimal value to generate smaller clusters. The final deepSplit of 4 was chosen and resulted in 55 modules with average size of 235 genes. Those identified functional modules are illustrated with different colours on the bottom of the Fig. 2A. The relationship between modules were further summarized by eigenvalue “eigengene”. Eigengenes are defined as the first left singular vector of the expression matrix for each identified functional module. Therefore, the eigengenes represent the expression profile with weighted genes for each module (Langfelder & Horvath, 2007).

Figure 2 The WGCNA analysis on the top 5,000 genes with most variation across 92 tissues in Bos taurus.

(A) Functional modules are illustrated with different colours. The parameter deepslip = 4 is set in WGCNA analysis, which providing a high sensitivity to cluster splitting. We additionally required each gene module with 30 or more genes. In total, 4,950 genes were grouped into 56 modules which showed with various colours. The top five modules ordered by number of genes were: turquoise with 212 genes; blue with 201 genes; brown with 187 genes; yellow with 162 genes; and green with 155 genes. The grey colour in the left of the figure represents the 50 genes not associated with any module. (B) The relationship tree for all the modules is presented and the top five modules marked in the corresponding number.

Pathway enrichment analysis and network analysis

We performed pathway enrichment analysis on those genes of interest by using functional enrichment tools in BGD (Elsik et al., 2016). This online tool includes enrichment in predefined pathways from KEGG and Gene Ontology. In these functional enrichment analysis, all the Ensembl Genes (24616) was used as background. The reconstructed co-expression network from WGCNA was visualized using the Cytoscape (version 3.4). The topological centrality analysis was performed by using NetworkAnalyzer in Cytoscape (Shannon et al., 2003). We used degree to represent the number of connections for each node in a network, and the shortest path represented by the least number of steps from one node to another (Barabasi & Oltvai, 2004). By using the sub-network extraction algorithm described in our previous study (Zhao, Liu & O’Mara, 2015), we built a sub-network to link the 340 DNA binding genes with the other cattle genes. Based on the graph theory, the searching for the sub-network for a set of genes in a graph is the Steiner tree problem. The algorithm assumed that all the 340 genes had zero cost for generality. Initially each gene formed a tree itself. Then, a greedy searching algorithm was used to iteratively merge the trees into larger trees based on our co-expression data until there was only one tree.

Results

Functional module identification on co-expression network using WGCNA and functional enrichment analyses for the genes in the top five modules

To determine the similarity between genes, the WGCNA consider not only the co-expression coefficients between genes, but also the content of co-expressed gene partners. To this aim, a topological overlap matrix (TOM) was calculated based on the adjacent coefficient and how many shared “friends” (interacting partners) between any pairs of co-expressed genes. In this way, all the edges between co-expressed genes were weighted by TOM ranging from 0 to 1, which represent the strength of the co-expression between the two genes. To identify the clustered co-expressed genes with specific functions, we further conducted module identification using agglomerative hierarchical clustering based TOM (Fig. 2A). Since it was hard to associate small number of genes to specific biological function, we required any functional modules with at least 10 genes.

To validate the potential functions for the modules, we focused on the top five modules with most genes (Table S2). Pathway and gene ontology (GO) enrichment analysis of the chosen modules were performed with BovineMine of BGD. Table 1 shows functionally enriched pathways obtained from BovineMine by setting adjusted P-value <0.05. We found enriched pathways only for module 1 and module 2. The genes in module 1 were identified as associated with metabolic pathways: there are three genes related to isoleucine degradation. A previous carbon-14 labelling experiment showed that the degradation of valine, leucine, and isoleucine represent a potential source of energy to the mammary gland as well as a source of carbon and alpha-amino nitrogen for the synthesis of nonessential amino acids (Wohlt et al., 1977). The genes from module 2 have extensive roles in extracellular processing and are associated with 15 pathways (Table 1). These pathways are known to be key components in the extracellular signaling system that involve collagen formation and degradation, glycosaminoglycan metabolism and axon guidance (Table 1).

Table 1 The enriched KEGG pathways for the genes in module 1 and 2 from WGCNA analysis.

Pathway	# of genes	Q-value*	
Module 1			
Metabolism	43	6.26E−07	
Isoleucine degradation	3	0.04218	
Module 2			
Collagen formation	14	4.54E−12	
Extracellular matrix organization	21	4.92E−12	
Collagen biosynthesis and modifying enzymes	13	1.54E−11	
ECM proteoglycans	11	2.85E−10	
Collagen degradation	10	7.38E−09	
Assembly of collagen fibrils and other multimeric structures	9	3.76E−08	
Degradation of the extracellular matrix	12	5.32E−08	
Integrin cell surface interactions	11	2.07E−07	
NCAM1 interactions	6	8.55E−06	
Glycosaminoglycan metabolism	9	0.00409	
MET activates PTK2 signaling	5	0.00967	
Cooperation of PDCL (PhLP1) and TRiC/CCT in G-protein beta folding	5	0.01919	
Non-integrin membrane-ECM interactions	5	0.02361	
Axon guidance	14	0.04552	
Notes.

* Q-values: the raw P-values of the hypergeometric test were corrected by Benjamini–Hochberg multiple testing correction.

By using the GO enrichment analysis, we further discovered more functional features for the five modules (Table 2). Those genes in module 1 (M1) are mainly metabolism related pathways (all adjusted P-values < 0.05). The components of module 2 (M2) are associated with extracellular structure organization and protein hetero-trimerization and trimerization (adjusted P-values < 0.05). The genes in module 3 (M3) use a microtubule cytoskeleton to organize cell projection (all adjusted P-values < 0.05). The module 4 (M4) is mainly related to pigment cell differentiation and its regulation (two adjusted P-values < 0.05). The genes in module 5 (M5) are enriched for the development of sertoli cells (adjusted P-values < 0.05), which are essential for spermatogenesis. Based on Pearson correlation coefficients, we further explored the relationship between modules. The module eigengenes are further calculated, which provides quantitative assessments for the similarity between the modules (Table S3). As shown in Fig. 2B, the top five modules are not clustered together which implies that they have different functions. Combined with our functional results from KEGG pathway and GO, we concluded that the top five modules have distinct and independent functions at the cellular level.

Table 2 The enriched biological processes GO terms for the genes in the top five modules from WGCNA analysis.

Modules	GO: biological process	Q-values*	
M1	Small molecule metabolic process	0.000971	
M1	Carboxylic acid metabolic process	0.00332	
M1	Oxoacid metabolic process	0.003628	
M1	Organic acid metabolic process	0.005205	
M1	Single-organism metabolic process	0.041382	
M2	Extracellular matrix organization	0.000392	
M2	Extracellular structure organization	0.000427	
M2	Protein heterotrimerization	0.000438	
M2	Collagen fibril organization	0.000636	
M2	Protein trimerization	0.004188	
M3	Cell projection organization	0.013119	
M3	Microtubule cytoskeleton organization	0.028215	
M3	Microtubule-based process	0.045173	
M3	Nervous system development	0.04747	
M4	Pigment cell differentiation	0.006709	
M4	Regulation of pigment cell differentiation	0.008956	
M4	Developmental pigmentation	0.024965	
M4	Melanocyte differentiation	0.026407	
M5	Sertoli cell development	0.00372	
Notes.

* Q-values: the raw P-values of the hypergeometric test were corrected by Benjamini–Hochberg multiple testing correction.

The hub genes in a co-expression based interactome with manageable size

In contrast to the correlation-based network reconstruction, WGCNA considered not only the expression correlation between two genes but also how many co-expressing genes were shared. In WGCNA, the weighted measure TOM was used to reflect the strength of the co-expression between the two genes and ranged from 0 to 1. In theory, the reconstructed network comprised all the 5,000 genes based on the TOM of >0. However, the resulting network is too large for functional genomics analysis. Since our aim was to build a comprehensive interactome covering as many genes with variant expression as possible, we defined three set of the co-expression gene network by using different TOM thresholds. For a TOM >0.01, the resulting co-expression based interactome comprised 4,995 genes with 1,538,522 significant co-expression pairs, which is too huge to visualize.

With a TOM >0.1, the interactome comprised 4,403 genes with 72,306 significant co-expression pairs (Table S4). For TOM scores greater than >0.3, there were 2,119 significant co-expression pairs and 1,045 genes (Table S4). Although the co-expression pairs were substantially reduced, the network was not fully connected (Fig. S1). Therefore, to visualize the entire network, we used a TOM score >0.1 which covered about 90% genes in the 5,000 genes (Fig. 3A). Since the network is still too large to obtain detail, we performed the topologic analysis to reveal the network structure. The diameter of the network is 11 and the average number of neighbors is 32.844. Further network topological analysis revealed that most genes in the reconstructed co-expression network are closely connected. In detail, we found that the probability P (k) for genes with other k co-expressed genes could be fitted to a power law distribution (P(k) ∼ kβ). The estimated β is 1.368 (Fig. 3B), which indicate this co-expression network are more closely connected compared to published human protein-protein interaction network with estimated β value of 2.9 (Jin et al., 2013). By further analysis the shortest pathways between all the co-expressed genes, we found the majority of the genes could connected with other genes by co-expressing with three or four more genes (Fig. 3C).

Figure 3 The co-expression network and gene ontology analysis of 340 genes with 100 or more connections.

(A) Co-expression network from WGCNA based on the TOM greater than 0.1; (B) the distribution of the number of connections for all the nodes in the network; and (C) short path length frequency for the network. The scatterplot (D) shows the gene ontology (GO) cluster representatives for the 340 genes in a two-dimensional space derived by applying multidimensional scaling to a matrix of the GO terms semantic similarities. Bubble colour indicates the corrected P-value of the GO term.

In addition, our reconstructed network also helped to identify a number of genes with hundreds of co-expressed genes. In general, these potential hub genes may have central roles for signaling transduction or metabolic transformation. In total, we identified 340 genes with 100 or more co-expressed genes (Table S5) and these genes are involved in fundamental processes: ribonucleotide binding (adjusted P-value = 1.253E−2, 54 genes); RNA binding (54 genes, adjusted P-value = 2.219E−3); RNA polymerase binding (6 genes, adjusted P-value = 4.696E−3); and cyclin-dependent protein kinase (5 genes, adjusted P-value = 1.440E−2). Additionally, there are 20 ATPases (adjusted P-value = 8.199E−3), which may indicate the importance of ATPases in the maintenance of metabolite homeostasis in Bos taurus.

Using the number of connections is the most common way to identify the key genes with important functions (Zhao & Qu, 2009). Interestingly, we identified API5 (apoptosis inhibitor 5) as the gene with highest degree (number of connection = 279). This apoptosis inhibitory protein often prevents apoptosis after growth factor deprivation in humans (Han et al., 2012). As one of the genes with most co-expressed gene partners, API5 may have critical functions in the cattle development and association with complex genetic traits. Another promising gene is FBXO11 with hundreds of co-expressed genes in Bos taurus genome (Table S5). As one of gene member of the F-box protein family, FBXO11 was functioned as a suppressor of p53 function by post-translational modification (Abida et al., 2007). In summary, our reconstructed co-expression network across 92 tissue samples may provide unexplored functional clues for many of the genes with a large number of connections in Bos taurus.

A gene-gene interaction sub-network related to DNA binding

To demonstrate the application of our reconstructed interactome, we downloaded 614 DNA binding genes from BGD (Table S6). Then, we connected these genes to form a functional network using the method implemented in our previous studies (Zhao, Chen & Qu, 2016a). The resulted sub-network contained 132 genes and 251 interactions (Fig. 4A, Table S7). In total, there were 104 genes from our original DNA binding genes, and 28 genes functioned as linker genes to fully connect the DNA binding genes. The number of connections of all genes followed a power law distribution P(k) ∼ k−b, where b is estimated as 1.388 (Fig. 4B) comparing to 1.368 (Fig. 3B). Although only 17% of the 614 DNA binding genes are co-expressed, they all formed highly modular structures, which implies coordination in DNA binding-related gene regulation. For example, we found 39 genes were involved in regulation of transcription from RNA polymerase II promoter (adjusted P-value = 2.04E−11). Similarly, there are 36 genes associated with “positive regulation of gene expression” (adjusted P-value = 2.04E−11) and 26 genes associated with “negative regulation of gene expression” (adjusted P-value = 2.43E−5). Taken together, the competitive regulation may be associated with RNA polymerase II promoter regions. With regard to the 28 linker genes, we found only three genes (AGO4, CAPRIN1, CNOT3) localized to “P-body” (GO:0000932, adjusted P-value = 0.03) and two genes (AXIN1 and CALCOCO1) that have “armadillo repeat domain binding” (GO:0070016, adjusted P-value = 3.24E−2). Although the majority are not statistically over-represented in any functional modules, their strong co-expression with hundreds of DNA binding regulators may imply their important role in cellular processes.

Figure 4 The sub-network for the DNA binding genes in Bos taurus.

(A) the sub-network extracted for DNA binding genes; (B) the distribution of the number of connections for all the nodes in the network; (C) the short path length frequency for the network.

In summary, by applying the sub-network extraction to the DNA binding genes in Bos taurus, we successfully identified a sub-network with hundreds of DNA binding genes and a number of relevant novel genes. This demonstrated that the use of our reconstructed co-expression interactome is a powerful approach to cluster genes with similar function for network-based data mining in cattle genetics and genomics studies in general.

Discussion and Conclusion

The cellular machines can be viewed as the product of thousands of proteins necessary to maintain cellular signalling and respond to extracellular stimulation. The genome-wide gene expression is coordinated in part through networks of protein–protein interactions that assemble functionally related proteins into complexes and organelles. Understanding the architecture of the Bos taurus transcriptome will improve our knowledge of cellular, structural and molecular mechanisms. For instance, those co-expressed genes may have similar biological functions. Also, this co-expression information could be used for elucidating how genome variation and expression contributes to the cattle breeding. Here we present a comprehensive co-expression based interactome in Bos taurus. However, it should be noted that the data used in this study are mainly for the mature mRNA without considering any post-transcriptional regulations. Therefore, it should be cautious for the data interpretation with potential post-transcriptional regulatory mechanisms.

By using robust co-expression analysis, we characterized a number of interesting genes for further investigation that formed tightly interconnected cluster in our co-expression network. Our further topological analysis revealed 340 highly-connected genes with 100 or more connections that may act as important links in various biological processes. For example, FBXO11 was identified to play a role in the p53 pathway. Combined with the results from the enrichment analysis of ribonucleotide binding, this gene may be one of the fundamental regulators involved in the suppression of p53 function. The p53 pathway was not only associated with bovine virus-induced leukemogenesis in cattle but is also important in human cancer (Zhao et al., 2016b). Therefore, the identification of p53 inhibitor, FBXO11, as a hub gene may provide a feasible approach for the design of molecular inhibitors to prevent p53-related diseases in cattle. Another interesting gene that shows a large connection in Bos taurus co-expression network is API5, an apoptosis inhibitor that is involved in the fibroblast growth factor binding. Since cell apoptosis has an important role in vitro-produced beef cattle embryos (Nkadimeng et al., 2016), our result may offer a number of new genes for identifying novel mechanisms of vitro-produced embryos in Bos taurus.

Our additional module analysis identified 55 highly-connected functional modules representing diverse cellular activities. By focusing on the top five modules with the largest number of genes, we characterized some important functions for these modules. For example, there are three genes (BCKDHA, ETFB, and PHLDB2) involving isoleucine degradation in module 1. More interestingly, the biochemical intermediates and final products from the isoleucine degradation pathway are the potential energy source for the mammary gland in cattle (Wohlt et al., 1977).

Moreover, our reconstructed network will serve as a basis for network-based mining as exemplified by the identified sub-network related to DNA binding genes in Bos taurus. This work highlights the importance of a systems biology approach to study largely unexplored transcriptomes by analysing the inherent modularity of the co-expression network concerned with the majority tissue samples. In conclusion, we performed the first systematically co-expression analysis on thousands of genes in Bos taurus genome across 92 tissues. The resulted co-expression pairs connected thousands of genes with similar functions and formed the first cattle interactome for large scale systems biology-based data mining.

Supplemental Information

Figure S1 The network based on the weight threshold of 0.3

Click here for additional data file.

Table S1 The expression profile for the top 5,000 most variant genes across 92 tissue samples

Click here for additional data file.

Table S2 The top five gene modules with most genes in WGCNA analysis

Click here for additional data file.

Table S3 The eigengenes for the gene modules from WGCNA analysis

Click here for additional data file.

Table S4 The edge information for the networks based on the weight threshold of 0.1 and 0.3

Click here for additional data file.

Table S5 The number of connections for all the genes in the co-expression network from WGCNA

Click here for additional data file.

Table S6 The gene related to DNA binding in Bos Taurus

Click here for additional data file.

Table S7 The gene types for the extracted sub-network related to DNA binding

Click here for additional data file.

Additional Information and Declarations

Competing Interests

Author Contributions

Data Availability

Dr. Min Zhao is an Academic Editor for PeerJ.

Yan Chen and Yining Liu conceived and designed the experiments, analyzed the data, wrote the paper, prepared figures and/or tables, reviewed drafts of the paper.

Min Du conceived and designed the experiments, reviewed drafts of the paper.

Wengang Zhang performed the experiments, prepared figures and/or tables, reviewed drafts of the paper.

Ling Xu reviewed drafts of the paper.

Xue Gao and Lupei Zhang performed the experiments, reviewed drafts of the paper.

Huijiang Gao and Lingyang Xu analyzed the data, reviewed drafts of the paper.

Junya Li conceived and designed the experiments, contributed reagents/materials/analysis tools, wrote the paper, reviewed drafts of the paper.

Min Zhao conceived and designed the experiments, analyzed the data, contributed reagents/materials/analysis tools, wrote the paper, prepared figures and/or tables, reviewed drafts of the paper.

The following information was supplied regarding data availability:

The data used in this study found in the Tables S1–S7.

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
