# Peer review of "Constructing a comprehensive gene co-expression based interactome in Bos taurus"

_PeerJ, doi:10.7717/peerj.4107_

## Round 0.1 · original submission · Major Revisions

Both reviewers have provided specific comments and suggestions that must be addressed during revision. Besides comments on missing details about statistical analyses of differential expression and need to clarify several points, the issue of "validation" of some of the findings is important.

Reviewer 1 ·

Basic reporting

The article is acceptable in this area

Experimental design

1- The authors claimed that they constructed the first multi-tissue gene co-expression network in cattle (Lines 1,23,68,162,289,231 and 323), which is not true. The first large-scale multi-tissue gene co-expression network in cattle has been already constructed by Beiki et al (2016).
2- Combining a large number of experiments into a single robust analysis will minimize the effects of variables that can plague individual experiments. In contrast, the authors just used gene expression information from single animal (one sample per tissue) to construct the cattle multi-tissue network. Why not use all available gene expression data for this species?
3- It is well known that unwanted noise and un-modeled artifacts such as batch effects can dramatically reduce the accuracy of statistical inferences in high-throughput experiments. Unfortunately, the authors did not adjust the data for these sources of noise, which may have biased all the results of this study.
4- The version of the reference genome used in this study need to be added in the method section (line 80).
5- What’s the rationale behind the selection of the most variable genes? Filtering genes based on their variability will result in information loss during network construction. For example, while transcription factors may have very subtle expression changes between different biological conditions, the may have a very large phenotypic impact. Variation filtering can easily lose this useful information.
Also, line 141-143, the authors mentioned that because the large number of genes were not differentially expressed between samples (tissues?), they decided to filter genes based on their variability. Differential expression analysis results are highly dependent on the number of samples used in the study. How many samples used per tissue in this study? If only one, how can differential expression analysis be performed based on a single sample per tissue?
6- Eigen value definition needs to be corrected on line 110. In WGCNA, Eigenvalues are calculated based on Singular Value Decomposition (SVD) not PCA and defined as the first left singular vector (referred to as the eigenvector) that explains the maximum amount of variation in the expression matrix.
7- Instead of the description of network topology based of different network statistics specifically developed for WGCNA networks (network heterogeneity, network density, network centralization…) why did the authors use topological centrality analysis in Cytoscape (line 119)?
8- What is the benefit of the shortest path analysis (line 121) while TOM based neighbor analysis can be easily conducted on the data?
9- the Results and methods sections are not fully separated in the manuscript. For example, lines 128-177 described the method used in the study while included in the results section.
10- The reference gene set used in the functional enrichment analysis (KEGG pathway and GO term analysis) was not included in the method section.
11- The significant level of connection between genes needs to be selected based on the TOM values distribution. What basis did the authors use - 0.01 as the significant TOM connection strength (line 215)?
12- The sentence at line 130-131 needs to be re-written as it unclear.

Validity of the findings

I have enough concerns with the methods used that I am not confident in the validity of the finding of this project.

·

Basic reporting

The paper from Chen et al. is overall written well with however some typos and/or unclear sentences (e.g., L105-107, confusing the “as more gene as possible”; L115: “interested genes” should be “gene of interest”) throughout the whole manuscript.
The authors are right in pointing out the importance of having an interactome in bovine. The author’s hypothesis was “that the complex genetic traits related to cattle production are reflected by the perturbation of gene-gene co-expressing networks”. The method used (Weighted Gene Co-Expression Network Analysis) in the transcriptome of 92 tissues of bovine without any type of animal production data associated with them does not seem to be able to attempt to demonstrate the hypothesis. I suggest the authors to change the hypothesis and, maybe, provide a statement of need instead of a hypothesis, based on the fact that such large co-expression database is missing.
What provided in this paper is a co-expression dataset using transcripts. The transcripts are, in general, mature mRNA (or other types of RNA); thus, the coexpression analysis using the transcriptome somewhat disregards post-transcriptional regulations and, therefore, do not really reflect the real co-regulation at DNA level. I would like the authors to consider this point.
It is unclear to the reviewer why the authors had selected the genes with the highest variability between tissues considering that the purpose was to study co-expressed genes.

Experimental design

No comments

Validity of the findings

The authors did not provide any validation of their findings. Because they purpose was to find co-expressed genes I suggest to check if any of the networks detected present enrichment of up-stream regulators. If the genes are really co-expressed they should have commons up-stream regulators.

Additional comments

L119: unclear what is “degree”. Please, clarify
L134: unclear the word “mature”
L142: how the statistical analysis was run to identify differentially expressed genes?
L171: unclear the word “friends”
L210: the method used can aid in identify co-expressed genes but cannot really identify communication between genes. I suggest changing in “coexpression”
L290-292: unclear how the method used by the authors can characterize subcellular localization and complex formation of genes products.

---

## Round 0.2 · Minor Revisions

I appreciate you taking into account all the initial comments from both reviewers. There are a few more minor comments that need to be addressed.

Reviewer 1 ·

Basic reporting

Acceptable

Experimental design

Acceptable

Validity of the findings

acceptable

Additional comments

This statement in the abstract “By using robust WGCNA (Weighted Gene Correlation Network Analysis), we described the gene co-expression network of 13,405 protein-coding genes from the cattle genome.” Is not correct given this statement “Using 19,064 genes with expression values, we ran a quality control step and removed those genes without expression values in more than half of 92 tissues. This provided a list of 13,405 genes with expression across 92 tissues. However, a large number of these genes were not differentially expressed between samples. Therefore, the data set with 13,405 gene expression was processed further by focusing on the 5,000 most variant genes (Table S1). The remaining 8,405 genes, which showed no or very low changes in expression between samples, were not used for WGCNA analysis.” In the materials and methods section. If I am understanding correctly, the network is for only 5000 genes. If this is correct, the abstract needs to be corrected.



I am confused by this statement “However, a large number of these genes were not differentially expressed between samples”. Based on the authors response, no differential gene expression analysis was conducted. Given that how can this statement be made?

---

## Round 0.3 · accepted · Accept

Authors did a good job taking into the account the last round of reviews.